# Comparison between Multiple Doses and Single-Dose Steroids in Preventing the Incidence of Reintubation after Extubation among Critically Ill Patients: A Network Meta-Analysis

**DOI:** 10.3390/jcm10132900

**Published:** 2021-06-29

**Authors:** Chiwon Ahn, Min Kyun Na, Kyu-Sun Choi, Tae Ho Lim, Bo-Hyoung Jang, Wonhee Kim, Youngsuk Cho, Hyungoo Shin, Jae Guk Kim, Juncheol Lee

**Affiliations:** 1Department of Emergency Medicine, College of Medicine, Chung-Ang University, Seoul 06974, Korea; cahn@cau.ac.kr; 2Department of Neurosurgery, College of Medicine, Hanyang University, Seoul 04763, Korea; mavmav@hanmail.net; 3Department of Emergency Medicine, College of Medicine, Hanyang University, Seoul 04763, Korea; erthim@gmail.com (T.H.L.); seodtst@gmail.com (H.S.); jclee0221@gmail.com (J.L.); 4Department of Preventive Medicine, College of Korean Medicine, Kyung Hee University, Seoul 02453, Korea; bhjang@khu.ac.kr; 5Department of Emergency Medicine, College of Medicine, Hallym University, Chuncheon 24252, Korea; wonsee02@daum.net (W.K.); faith2love@hanmail.net (Y.C.); gallion01@daum.net (J.G.K.)

**Keywords:** steroid, planned extubation, reintubation, network meta-analysis

## Abstract

This study aimed to determine the frequency of prophylactic steroid administration to prevent reintubation after extubation in critically ill patients. We systematically searched MEDLINE, Embase and Cochrane Library for studies regarding the preventive use of multiple doses or single-dose steroids prior to extubation on July 2020 and conducted a network meta-analysis (NMA) to compare these interventions. To assess the risk of bias of each included study, version 2 of the Cochrane risk-of-bias tool for randomized trials was used. Nine randomized control trials comprising 2098 patients with comparisons of the three interventions were included. Use of multiple doses and single doses of intravenous steroids administration showed a significantly lower rate of reintubation compared with placebo (odds ratio [OR]: 0.43, 95% confidence interval [CI]: 0.25–0.72; OR: 0.31, 95% CI: 0.14–0.69). However, the comparison between multiple doses and single doses showed no significant differences (OR: 1.22, 95% CI: 0.32–4.74). According to the surface under the cumulative ranking curve statistic, the treatments should be ranked as follows: single dose (87.1%), high dose (62.8%) and placebo (0.1%). This NMA showed that the multiple doses were not statistically superior to the single dose in lowering the incidence of reintubation after extubation in critically ill patients. Therefore, use of a single-dose steroid can reduce the incidence of reintubation.

## 1. Introduction

Reintubation after extubation failure causes complications such as cardiovascular failure or ventilator-associated pneumonia and is associated with an increased mortality rate [1,2,3]. Extubation should not be considered as a simple reversed process of intubation. Even if the patient’s medical condition is stable and no potential complications occurred, such as post-extubation stridor, laryngeal edema or reintubation, the need for extubation should be planned on the day that the patient was intubated [4,5,6]. The current guideline suggests that the prophylactic use of steroids prior to extubation is recommended as it is effective in reducing inflammatory airway edema, which can cause direct airway injury [4,7]. A recent meta-analysis also showed that the use of steroids is effective in reducing post-extubation stridor and reducing the incidence of reintubation after extubation [8].

The difficult airway society guidelines suggest that all steroids equivalent to 100 mg of hydrocortisone (HC) should be administered every 6 h [4]. In addition, Lin et al. (2016) showed that the administration of multiple doses is effective in reducing post-extubation airway obstruction than the use of a single dose [9]. However, despite the suggested steroid dose and administration method, various steroid regimens have been used in previous studies. The diversity in the steroid dose and administration methods has unsettled practicing clinicians.

We performed a network meta-analysis (NMA), which allows a coherent analysis of all randomized controlled trials (RCTs), in order to evaluate the association between the intravenous (IV) administration of prophylactic steroids and the reintubation rate after extubation in critically ill patients. Additionally, we analyzed the efficacy of various steroid doses (multiple doses, single dose and placebo) with IV administration, by integrating all available direct and indirect evidence in the NMA.

## 2. Materials and Methods

This NMA was performed in accordance with the Preferred Reporting Items for Systematic Reviews and Meta-Analyses statement for reporting network meta-analyses [10].

### 2.1. Search Strategy

MEDLINE, Embase and Cochrane Library were systematically searched by two independent reviewers (Ahn C and Na MK) for studies regarding the preventive use of multiple doses or single-dose steroids prior to extubation in critically ill medical patients from inception through July 2020. We searched without language restrictions. The detailed search strategy is presented in Appendix A. The reference lists of previous relevant studies and review articles were manually searched to identify other relevant literature.

### 2.2. Study Selection

After excluding duplicate records, two reviewers (Ahn C and Na MK) independently screened the title and abstract of all selected articles to assess for eligible studies. During the preliminary screening, the full text of eligible articles was reviewed to determine whether these studies met the inclusion or exclusion criteria. No restrictions were imposed on the study period or type of steroid. The third reviewer (Choi K-S) participated in the discussion to decide whether or not to include in the meta-analysis when there was a difference of opinion between the two authors on articles with the potential for inclusion.

### 2.3. Inclusion and Exclusion Criteria

(1) Studies conducted in adult critically ill medical patients (age ≥18 years) admitted in the intensive care unit who used steroids with IV administration to prevent complications after extubation, (2) studies that reported the outcomes: reintubation and (3) prospective RCTs were included in the analysis.

Meanwhile, (1) non-RCTs, including reviews, cohort studies and crossover studies; (2) studies conducted in post-surgical patients; (3) studies that enrolled patients who underwent an unplanned extubation; (4) studies that did not report the outcome of interest; and (5) conference abstracts without full-text manuscripts were excluded.

### 2.4. Groups of Comparison: Multiple Doses vs. Single Dose vs. Placebo

The multiple dose group referred to patients who received two or more steroid injections on a regular interval prior to extubation. The single-dose group referred to patients who received one steroid injection. The placebo group referred to patients who did not receive steroid injections.

### 2.5. Data Extraction

Two reviewers (Ahn C and Na MK) independently reviewed the full text of each included study and extracted the data using a standardized form. The abstracted data included the name of the author, publication year, duration of study, setting of the study, sample size, details of the population enrolled, type of IV steroid used, interval of IV steroid administration and equivalent dose of HC that can achieve a relative anti-inflammatory effect [1 mg methylprednisolone (MP) = 5 mg HC, 1 mg dexamethasone (DM) = 25 mg HC] (Glucocorticoid equivalents in Appendix A) [11]. When discrepancies occur between reviewers, a discussion was made to reach a consensus.

### 2.6. Quality Assessment

For the quality assessment of the included studies, version 2 of the Cochrane risk-of-bias tool for randomized trials (RoB 2) was used [12]. RoB 2.0 was divided into the following categories: “risk of bias arising from the randomization process,” “risk of bias due to deviations from the intended interventions,” “risk of bias due to missing outcome data,” “risk of bias in measurement of the outcome,” and “risk of bias in selection of the reported result.” Each subcategory was rated as follows: “yes,” “probably yes,” “no,” “probably no,” and “no information.” Next, using the evaluation result of each subcategory, the risk of bias was evaluated as low, high, or some concerns according to the evaluation algorithm suggested in RoB 2.0. Finally, the risk of bias was determined as low, high or some concerns according to the overall quality evaluation criteria presented in RoB 2.0. Disagreements between the two reviewers were resolved by discussion.

### 2.7. Statistical Analysis

The random effects NMA was performed using a frequentist framework to calculate the ORs for dichotomous outcomes and the corresponding 95% CIs. All statistical analyses were performed using the netmeta package in Stata 13.0 (Stata-Corp, College Station, TX, USA). A two-sided *p* value of less than 0.05 was considered significant. The homogeneity and consistency assumptions underlie the validity of evidence from NMA [13]. The inconsistencies between direct and indirect estimates in the entire network for each outcome were assessed locally with a loop-specific approach and globally with a design-by-treatment interaction model [14]. The treatment effects of various respiratory support methods were ranked according to the probabilities of leading to the best results based on the surface under the cumulative ranking (SUCRA) for each outcome [15]. The SUCRA value ranged from 0% to 100%; a higher SUCRA value indicated the effectiveness of the method [15]. Each comparison was conducted using Grading of Recommendations Assessment, Development and Evaluation (GRADE) analysis (GRADEpro Guideline Development Tool, McMaster University, USA) [16]. Finally, we inspected the funnel plot for presence of publication bias using the netfunnel package in Stata 13.0.

## 3. Results

### 3.1. Study Selection

After the online database search, 1914 relevant articles were found in MEDLINE, 1844 in Embase and 1871 in the Cochrane Library; meanwhile, two additional articles were found by manual searching [17,18]. A total of 3786 studies were identified after removal of duplicates and 65 potentially relevant articles were retrieved after a full-text review (Figure 1). The final nine RCTs selected met the eligibility criteria and included 2098 patients [9,17,18,19,20,21,22,23,24].

### 3.2. Study Characteristics

Of the nine studies, seven RCTs that were published in Asia and two were published in France (Table 1). The steroids used for treatment were MP, DM and HC. Regarding the methods of steroid administration, four studies used multiple doses, three studies used a single dose and two studies used both single and multiple doses. With regard to the use of multiple doses or single dose, the amount to be administered once was an HC equivalent dose of 100–250 mg. In Lin et al.’s study (2016), the two different amounts of steroids were used in one comparison because both were administrated in multiple doses [9].

### 3.3. Risk of Bias

With regard to the overall risk of bias for each included study, five studies had a low risk of bias, while four studies had some risk of bias. In the detailed assessment of all subcategories, the “risk of bias arising from the randomization process” of four studies were evaluated as some concerns risk of bias because the detailed descriptions of the randomization process were omitted or the baseline difference between intervention and control was shown in Appendix A.

### 3.4. Quality of Evidence

We assessed the quality of evidence from each comparison for reintubation using the GRADEpro Guideline Development Tool. The GRADEpro tool revealed a high level of evidence between multiple doses and placebo and a high level of evidence between single doses and placebo. However, the tool showed a moderate level between multiple doses and placebo due to imprecision (Table 2, Appendix A).

### 3.5. Main Analysis: Pairwise Meta-Analysis

Each comparison included multiple doses, single dose and placebo. There were six direct comparisons for multiple doses and placebo, five for single dose and placebo and two for multiple doses and a single dose (Table 2, Appendix A). The multiple doses showed a significantly lower rate of reintubation compared with placebo (odds ratio (OR): 0.43, 95% confidence interval (CI): 0.25–0.72) and the single dose showed a significantly lower rate of reintubation compared with placebo (OR: 0.31, 95% CI: 0.14–0.69). However, the comparison between multiple doses and single doses showed no significant differences (OR: 1.22, 95% CI: 0.32–4.74) (Table 2, Appendix A). There was no significant heterogeneity among the included studies within each comparison.

### 3.6. Main Analysis: Network Meta-Analysis

Figure 2 shows the forest plot of the overall comparison. The inconsistency test at the global and local levels indicated no significant inconsistency (global level: *P* = 0.9942; local level: *P* = 0.995, 0.686 and 0.824; Appendix A); there is no problem with accepting the consistency model. Indirect evidence showed that multiple doses or a single dose likely decreased the reintubation rate compared with placebo (Table 3). According to the surface under the cumulative ranking curve (SUCRA) statistic, which calculated the probability of each treatment, the treatment should be ranked in the following order: single dose (87.1%), multiple doses (62.8%) and placebo (0.1%) (Table 4). Results of the statistical analysis showed that the single dose is superior to the multiple doses in terms of treatment effect.

### 3.7. Publication Bias

To assess for publication bias, we created a funnel plot (Appendix A). Overall, the studies showed 95% CIs. In addition, the plot appeared symmetrical in shape.

## 4. Discussion

This NMA indicated the necessity of administering steroids in multiple doses or in a single dose to prevent the risk of reintubation after extubation in critically ill patients and assessed the outcome-specific certainty of evidence using the GRADE system compared with previous meta-analysis. The pairwise meta-analysis showed that there was no significant difference between multiple doses and single-dose steroids in terms of reducing the risk of reintubation; however, the NMA showed that the use of a single dose was statistically superior to the use of multiple doses. Additionally, both multiple doses and single-dose steroids have shown significant superiority in lowering the reintubation rate compared with placebo in several studies.

The results of this study were inconsistent with those of previous studies, which reported that multiple doses are more effective than single-dose steroids [4,9]. The duration of action of steroids is at least 8 h; MP, an intermediate-acting steroid, 12–36 h; and DM, a long-acting steroid, 36–72 h [11]. In the included studies, multiple doses were administered every 4–6 h. Considering the decrease in body distribution over time and the increase of drug concentration in the blood, multiple doses are a relatively high dose and the duration of action is longer than that of a single dose. However, the evidence that multiple doses of steroids can lower reintubation than single doses remained unclear. The dosage of steroids and duration of use are important independent risk factors for the occurrence of side effects [25]. If multiple doses of steroids do not significantly lower the occurrence of reintubation compared with a single dose, this dosing method should not be used as it is cumbersome, time consuming and requires more manpower.

The steroids used before extubation are short acting and are administered in smaller doses; to date, no study has reported the direct side effects of steroid use during extubation. Even if side effects occur, they are reversible due to short-term use (<24 h) and more likely to be the effect of prolonged use of steroids in critically ill patients rather than the side effect of the use of prophylactic steroids [26,27]. Therefore, clinicians might consider the use of steroids to prevent the occurrence of complications after extubation based on the patient’s condition and environment.

The last study included in the analysis was conducted in 2016 [9]; no RCT has reported the use of steroids to prevent reintubation after extubation. In recent years, noninvasive ventilation (NIV) and high-flow oxygen therapy (HFOT) after extubation to prevent reintubation are used as alternative respiratory supports [28,29,30]. The latest NMA reported that NIV is the most effective respiratory support method for preventing reintubation after extubation [31]. None of the studies included in this meta-analysis used alternative respiratory support after extubation. Steroids used prior to planned extubation and NIV and HFOT used after extubation are not opposite treatments. Although no previous RCTs have used steroids and alternative respiratory support sequentially, these treatment methods can prevent the incidence of reintubation; however, further research is needed to confirm this finding.

This study has several limitations. First, the use of three types of steroids can cause bias. Although a standard steroid dose equivalent to the HC dose has been established to achieve the relative anti-inflammatory effect, the interpretation of results is limited due to the differences in the duration of action, time of onset and duration of potency for each steroid. Second, a slightly direct comparison was performed between multiple doses and single dose. In the network statistics, statistical analysis was performed by adding the values of direct and indirect comparisons; statistically, the consistency was satisfied in this study. However, due to the small number of studies, the results cannot be considered as clinically significant. Hence, additional NMA should be conducted in future studies. Third, the included studies were conducted in Asia, except for two study and the generalizability of the results was limited because the studies were only focused on a certain ethnic group and regional area.

In conclusion, we suggest the prophylactic use of steroids regardless of the dose method (multiple doses or single dose) to reduce reintubation after extubation in critically ill patients. Although multiple doses of prophylactic steroids can increase the blood concentration and retention time, its effect were not statistically superior to single-dose steroids in reducing the incidence of reintubation after extubation among critically ill patients and a single dose of steroid might be sufficient to reduce the incidence of reintubation. Therefore, a single dose can be used to reduce the rate of reintubation considering the patient’s condition and environment.

## Figures and Tables

**Figure 1 jcm-10-02900-f001:**
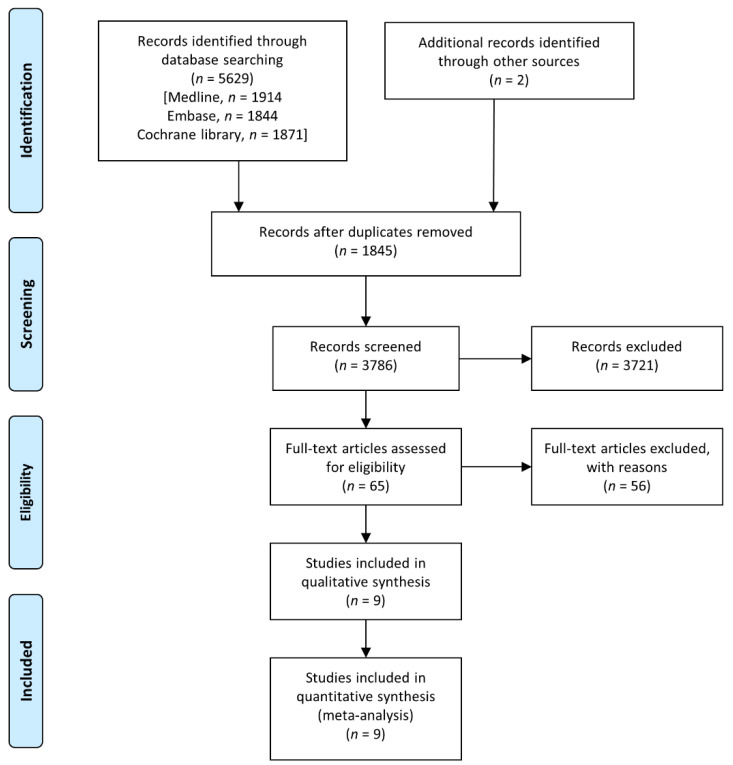
PRISMA flowchart of the study selection process.

**Figure 2 jcm-10-02900-f002:**
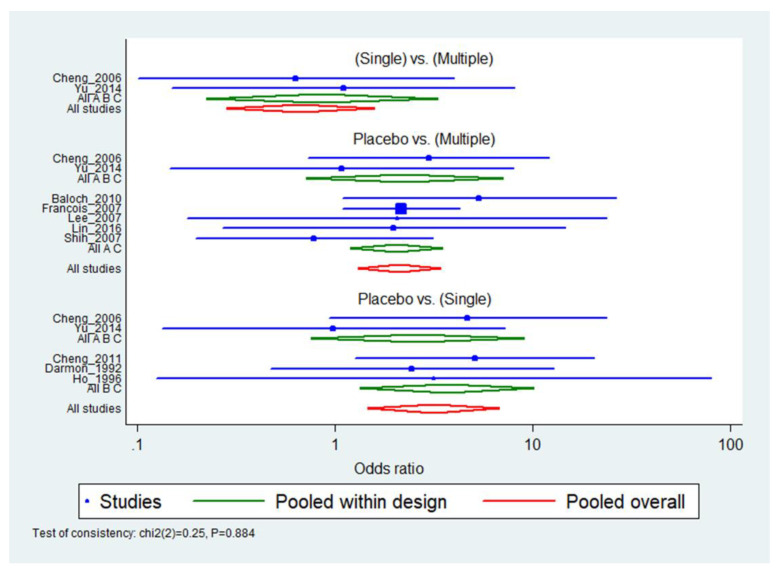
Forest plot of the network meta-analysis for reintubation rate.

**Table 1 jcm-10-02900-t001:** Baseline characteristics of included studies.

Author	Year of Publication	Region	Period	Population in Meta-Analysis	Age	Steroid	Dose Frequency	Equivalent Dose of Hydrocortisone *
Darmon et al. [19]	1992	France	1986–1987	664	53.16	DM	Single	200
Ho et al. [20]	1996	Taiwan	1990	77	62	HC	Single	100
Cheng et al. [21]	2006	Taiwan	2002–2004	128	66.12	MP	Multiple or Single	800 or 200
Francois et al. [22]	2007	France	2001–2002	698	66	MP	Multiple	400
Lee et al. [23]	2007	Taiwan	2004–2006	80	72.55	DM	Multiple	500
Baloch et al. [17]	2010	Pakistan	2006–2008	92	39.65	DM	Multiple	500
Cheng et al. [24]	2011	Taiwan	2005–2006	71	60.49	MP	Single	200
Yu et al. [18]	2014	China	2010–2013	162	67	DM	Multiple or Single	250 or 125
Lin et al. [9]	2016	Taiwan	2007–2010	126	74.09	DM	Multiple	1000 or 500

MP, methylprednisolone; DM, dexamethasone; HC, hydrocortisone. * 1 mg MP = 5 mg HC, 1 mg DM = 25 mg HC.

**Table 2 jcm-10-02900-t002:** Meta-analysis results for pairwise comparisons of steroid administration frequency.

Treatment 1	Treatment 2	No. of Studies	I^2^	OR (95% CI)	GRADE
Multiple doses	Placebo	6	0%	0.43 (0.25–0.72)	High
Single dose	Placebo	5	0%	0.31 (0.14–0.69)	High
Multiple doses	Single dose	2	0%	1.22 (0.32–4.74)	Moderate ^a^

I^2^, Higgins I square; OR, odds ratio.; ^a^ Quality of evidence for direct estimate rated down by one level for serious imprecision.

**Table 3 jcm-10-02900-t003:** League table for networks according to steroid administration frequency.

	Placebo	Multiple Doses	Single Dose
Placebo	-	0.42(0.25–0.69)	0.31 (0.14–0.67)
Multiple doses	2.41 (1.44–4.03)	-	0.75 (0.31–1.79)
Single dose	3.21 (1.49–6.92)	1.33 (0.56–3.20)	-

**Table 4 jcm-10-02900-t004:** Rank probability and surface under the cumulative ranking (SUCRA) curve result.

Rank	1	2	3	Mean Rank	SUCRA
Single dose	0.741	0.257	0.002	1.3	0.871
Multiple dose	0.259	0.741	0.001	1.7	0.628
Placebo	0.000	0.003	0.998	3.0	0.001

## Data Availability

The datasets generated during the current study are available from the corresponding author on reasonable request.

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
