# Peer review of "Comparison between Multiple Doses and Single-Dose Steroids in Preventing the Incidence of Reintubation after Extubation among Critically Ill Patients: A Network Meta-Analysis"

_jcm, 2021, doi:10.3390/jcm10132900_

Round 1

Reviewer 1 Report

Overall:

It is not an essentially new topic; the question of clinical relevance does arise here. The review itself was well written, but the question of the selection of the studies arises, especially in the partly inconsistent material & method part and in the results part (see recommendations). Furthermore, some citations and sources are unclear. Likewise, not all authors are listed in the contributions. There is still significant room for improvement here.

The statement that the single dose is superior is at least borderline, since the statistical differences are manageable and ultimately based on very few studies (Figure 2 supplementary; Are these two studies the two manually selected ones?)

Introduction:

Well written, short, on the point.

Materials and Methods:

Line 59/60 Citation 24: insufficient reference, no direct way to access the algorithm, please provide further information and cite the algorithm correctly

Line 71: how many articles were eligible? Percentage? This is a crucial fact as it limitates the study selection.

Line 73: please indicate the third reviewer

Line 99ff Citation 25-28: This type of citation is not permitted! One cannot cite an algorithm mentioned in another original paper. Please quote the underlying work and original publication.

Results:

Line 127: What exactly is manual search? Are the two studies included in the final nine studies? Then this would have to be indicated separately in the flow chart Figure 1. In addition, the question must be asked whether there is a conflict of interest here? Why weren't these studies captured by the search criteria and why were they included?

Line 135: There are 9 studies? (6+2 =8) Moreover, the sentence makes no sense at all: Of the six RCTs that were published in East Asia, two were published in France (Table 1)

Line 144, Table 1: How can you explain the exclusion of patients to your analysis? For example, Lee reported of 86 patients, in your table are only 80 mentioned. Please explain.

Discussion:

Consistent, well written

Author contribution:

Please clarify! There are ten authors listed, but in the contribution section not all of them are mentioned! Moreover, in the method section the information contradicts the author contribution in parts.

Reviewer 2 Report

Comments and Suggestion

The manuscript entitled “Comparison between multiple doses and single-dose steroids in preventing the incidence of reintubation after extubation among critically ill patients: A network meta-analysis”, presented by Chiwon Ahn et al. offers an opportunity to look into/understand how much evidence is available for using steroids  in preventing the incidence of reintubation among critically ill patients, also to determine whether a conclusion can be reached regarding to effect of giving steroids before scheduled extubation.

Scheduled extubation is a decision based on many criteria which patients must fulfilled before and to determine the most appropriate rescue plan for reintubation should extubation fail. Guidelines suggest using prophylactic steroids before extubation to avoid potential reintubation, however, even now, some date about this topic are still missing. Authors provide readers with some answers regarding the use of steroids in the prevention of incidents related to respiratory manipulation through the performed a network mata-analysis.

Some important points need clarification before.

Major

Materials and methods

  • Line 55 and 77 what kind of administration methods was considered?
  • Line 60 inadequate number of reference
  • Line 94 please provide reference[s], as well as methods of administration
  • Line 99 inadequate number of reference
  • Line 113 inadequate number of reference
  • Line 119 inadequate number of reference
  • Line 121 provide reference for GRADE analysis
  • Paragraph 2.2 provide if any limitation was used regarding language

Was any attempts made at collecting unpublished data?

  • Line 130 inadequate references 
  • In paragraphs 2.6 categories of bias was mentioned accept the “risk of biased arising from randomization process’. Please add.

There is no information about the reasons of exclusion so many studies.

Provide the descriptions of GRADE abbreviation as it is fist used in text, as well as for SUCRA

Results

  • Ad Table 1. Add references by the name of the first authors (firs column), it will help readers find the relevant  references.  In table the abbreviation NR is unnecessary as is not in the table/ text.
  • Table 2 – add description of I letter, and for T also.

Discussion

  • Line 205, 211, 225,227 inadequate number of reference
  • 194, 195 full description of GRADE abbreviation here is not necessary, as it should be done earlier in the text
  • Line 219- please provide references

References

  • All references throughout text are scrabbled up. Please put in correct order. Please check adequacy of the assigned references number in the main text and assign the correct one.

Supplementary file

  • S2. Add the number of comparisons on each arm.
  • Fig S5 - please named the figure of what was analyzed, as well as give what is A, B, C referred to.

Suggestion for authors:  

Overall, in this discussion, there is a lack of broad view of the problem of reintubation and  the interpretation of the results in the context of  these 9  studies included in metanalysis. Discuss the issue together with already existing metanalysis and, also, reviews (e.g.  Pluijms WA, van Mook WN, Wittekamp BH, Bergmans DC. Postextubation laryngeal edema and stridor resulting in respiratory failure in critically ill adult patients: updated review. Crit Care. 2015 Sep 23;19(1):295. doi: 10.1186/s13054-015-1018-2. PMID: 26395175; PMCID: PMC4580147.)  In the fourth paragraph authors mentioned  NIV and HFOT as “not opposite treatment”, that is true, since the mechanism of prevention for reintubation in the case of  alternative respiratory support is completely different. The main reason of steroids application is reducing anti-inflammatory, indirectly reducing  laryngeal oedema. The results of a meta-analysis are used to highlight the weakness of previous studies  and to recommend how to improve the design of future trials. Hither, there is no strong message how this network metanalysis could influences on existing guidelines, as it is one of main function of undertaking metanalysis, and to give possible  future study direction.

Minor

Introduction

  • line 39 and 39/40 the expression “the need for extubation should be planned” is repeated, please rephrase.
  • Line 45 – referenced guidelines [ref 4] do not suggest hydrocortisone, rather referred to all steroids, provided they are given in adequate doses, equivalent to 100 mg hydrocortisone. Please reword. Noteworthy, those guidelines relates to postoperative extubation issue, not critically ill patients
  • In line 48 - 50 phrase repetitions about “dose and administration methods” “previous studies”
  • In line 54/55 ‘in intubated ‘phrase is unnecessary

Materials and methods

  • Line 94 and 95 – this first appearance of steroids name abbreviations, please add full description. The equivalent dose was mentioned only for dexamethasone and methylprednisolone, although in search strategy (table 2) others name of steroid was searched as well. Perhaps, equivalency for all mentioned steroids should  be added.

Discussion

-line 213 second “is” in this line could be omitted

-line 242 rephrase, as the information provided is not compatible with the data  from table 1 (Taiwan- 5 studies, France -2 studies, China – 1 study, Pakistan - 1 study), maybe describe by using continents instead of  country

-line 250 rephrase as the provided information repeats what was wrote in first sentence. It could be enough mention  increasing risk of side effect parallel with high and multiple dose application of steroids.

Reviewer 3 Report

The use of the third reviewer in the choice of articles. Explain how your participation was and explanation of the existing disagreements for the use of this third reviewer

In the conclusions, they should give a slightly firmer conclusion regarding a dose and even opt for the most effective prophylactic in light of their observation.

Does your study justify the possibility of developing a health protocol for action to avoid reintubation?

Round 2

Reviewer 1 Report

The paper improved sufficiently. However, there are still some points left:

Line 121: typing error with SUCRA vs SCURA

Line 139: The sentence continues to make little sense. Probably the following is meant: Of the nine studies seven RCTS were published in Asia, and two were published in France (Table 1)

The inclusion of the two manually found studies must be explained in the paper itself, please explain this in the result Line 129 ff

Line 253: double “the”

Author Response

The paper improved sufficiently. However, there are still some points left:

Line 121: typing error with SUCRA vs SCURA

Line 139: The sentence continues to make little sense. Probably the following is meant: Of the nine studies seven RCTS were published in Asia, and two were published in France (Table 1)

The inclusion of the two manually found studies must be explained in the paper itself, please explain this in the result Line 129 ff

Line 253: double “the”

Response: Thanks for your pertinent comments.

We sincerely revised your comments.
